# Aquarius Sea Surface Salinity Gridding Method Based on Dual Quality–Distance Weighting

**Yanyan Li** [1,2], **Qing Dong** [1,*] **and Yongzheng Ren** [1,3]

1   Key Laboratory of Digital Earth Science, Institute of Remote Sensing and Digital Earth, Chinese Academy of Sciences, Beijing 100094, China; liyy@radi.ac.cn (Y.L.); renyz@radi.ac.cn (Y.R.)
2   University of the Chinese Academy of Sciences, Beijing 100049, China
3   Hainan Key Laboratory of Earth Observation, Sanya 572029, China
*   Correspondence: qdong@radi.ac.cn; Tel.: +86-10-8217-8121

**Abstract:** A new method for improving the accuracy of gridded sea surface salinity (SSS) fields is proposed in this paper. The method mainly focuses on dual quality–distance weighting of the Aquarius level 2 along-track SSS data according to quality flags, which represent nonnominal data conditions for measurements. In the weighting progress, 14 data conditions were considered, and their geospatial distributions and influences on the SSS were also visualized and evaluated. Three interpolation methods were employed, and weekly gridded SSS maps were produced for the period from September 2011 to May 2015. These maps were evaluated via comparisons with concurrent Argo buoy measurements. The results show that the proposed method improved the accuracy of the SSS fields by approximately 36% compared to the officially released weekly level 3 products and yielded root mean squared difference (RMSD), correlation and bias values of 0.19 psu, 0.98 and 0.01 psu, respectively. These findings indicate a significant improvement in the accuracy of the SSS fields and provide a better understanding of the influences of different conditions on salinity.

**Keywords:** Aquarius; sea surface salinity (SSS); dual weighting; quality

---

## 1. Introduction

Sea surface salinity (SSS) is one of the most important parameters in marine dynamics and is closely related to large-scale ocean circulation and climate change [1,2]. SSS values can be effectively measured based on data from the Aquarius satellite. Electromagnetic radiation emissions from the sea surface can be measured in the form of equivalent brightness temperatures in Kelvin by the Aquarius satellite and then converted into SSS after applying corrections for various geophysical effects. The Aquarius satellite was launched in June 2011 by the National Aeronautics and Space Administration (NASA) with the goal of providing monthly global SSS fields with a root mean squared difference (RMSD) of 0.2 psu over a spatial smoothing scale of 150 km [3,4]. The Aquarius satellite carries an L-band microwave radiometer (MWR) with a swath width of 390 km and an exact repeat orbit of 7 days.

The officially released standard Aquarius gridded SSS level 3 (L3) data products are generated from level 2 (L2) salinity data without any additional adjustments for climatology, reference model output or in situ data [5]. To assess and validate the accuracy of the Aquarius gridded SSS fields, tests have been conducted in both global oceans and regional basins [6–9]. Official Aquarius version 5.0 products have been published, and the accuracy of the monthly L3 1° SSS fields has been estimated via a triple point analysis using individual matchups among Aquarius data, Argo float data and Hybrid Coordinate Ocean Model (HYCOM) data [10,11]. The triple point analysis results showed that the monthly average RMSD of the Aquarius L3 field data was 0.128 psu, whereas the mean RMSD between the weekly

L3 analysis and the concurrent Argo float data was 0.247 psu. Garcia-Eidell et al. [12] studied the spatial and temporal distributions of SSS data at high latitudes through a comparative analysis of the Aquarius weekly L3 polar-gridded products (version 5.0). The RMSD was 0.465 psu relative to the Coriolis Ocean database for ReAnalysis (CORA) 5.0 data. The accuracy of gridded SSS fields directly affects their application. To improve the utility of the gridded SSS fields, Melnichenko et al. [13] used an optimal interpolation (OI) method to map the Aquarius L2 orbital SSS data. The orbital SSS data were first checked for quality according to quality flags. The weekly average OI SSS in the North Atlantic was calculated for the period from September 2011 to August 2013. The mean RMSD versus the Argo measurements was approximately 0.198. New weekly gridded SSS products on a nearly global 0.5° grid were also produced in 2016 [14]. The overall RMSD values versus independent thermosalinograph (TSG) salinity data and concurrent Argo observations were approximately 0.20 psu and 0.18 psu, respectively.

The "radiometer_flags" dataset is a key quality flag dataset associated with the Aquarius L2 products and represents nonnominal data conditions for radiometer measurements [15]. The Aquarius MWR does not measure the SSS directly; instead, it measures the microwave radiation signal emitted from the sea at 1–2 cm from the surface [9], which may be influenced by marine environmental factors. For example, the influence of the wind speed on the L-band brightness temperature (TB) is approximately 0.2–0.3 K for a 1 m/s change in wind speed according to previous studies [16,17]. Furthermore, when the incidence angle increases from 20° to 50°, the sensitivity of TB to the wind speed decreases from 0.14 K/ms$^{-1}$ to −0.03 K/ms$^{-1}$ for V polarization and increase from 0.29 K/ms$^{-1}$ to 0.36 K/ms$^{-1}$ for H polarization [18,19]. In addition, rain drops have a great impact on TB because they induce freshening and roughness effects on the sea surface [20]; especially under persistently rainy conditions with low winds, microwave measurements can strongly deviate from the real values [21]. Significant errors also exist near coastal areas because of the contamination from the nearby land. The results of Li et al. [22] show that the TB error is large when the distance to the coast (DC) is within 40 km and then decreases sharply from ~60 to ~4K, while with a further increase in DC from 40 to 400 km, the TB error decreases smoothly from ~4 to ~0.15 K. Therefore, the data quality is not constant under different conditions; consequently, orbital SSS data retrieved under different conditions should be treated differently. Although a number of studies have considered quality flags in the gridding procedure, these flags have been used only for data screening [5,14,23].

The present paper aims mainly to improve the accuracy of gridded SSS data at smaller spatial and temporal scales by means of a newly proposed method: dual quality–distance weighting of the orbital SSS data based on quality flags. In this method, the orbital SSS data are processed according to quality flags with the goal of maximizing the utilization of unflagged data. In this study, near-global weekly 0.25° × 0.25° gridded SSS fields were constructed for September 2011 to May 2015. Additionally, the constructed SSS fields were compared with concurrent individual Argo buoy observations to verify the accuracy of the SSS fields.

The remainder of the paper is organized as follows. Section 2 introduces the datasets used in this paper. Section 3 describes the method and procedures. The results and verification are presented in Section 4. Section 5 offers a discussion, and Section 6 concludes the paper.

## 2. Dataset

### 2.1. Aquarius SSS Data

The Aquarius SSS (version 5.0) product data used in the present study [5,15] were processed by the Aquarius Data Processing System (ADPS) and are distributed by NASA's Physical Oceanography Distributed Active Archive Center (PODAAC) of the Jet Propulsion Laboratory (JPL). The processed data are stored as a single physical hierarchical data format (HDF) file, which can be downloaded from https://oceandata.sci.gsfc.nasa.gov/Aquarius/.

The Aquarius SSS L2 (version 5.0) products from September 2011 to May 2015 are used in this paper. The L2 data are structured as a sequence of files, each corresponding to one orbit of Aquarius. The orbit has an exact repeat cycle of 103 orbits (~7 days). The Aquarius weekly gridded SSS L3 products (version 5.0) from September 2011 to May 2015 are also used in this paper for comparison with the results.

### 2.2. Argo Data

The Argo project provides in situ salinity profiles over the global ocean collected by over 3000 free-drifting profiling floats, which measure salinity and temperature from near the surface to 2000 dbar [24]. The Argo salinity measurements are accurate to better than 0.01 psu [2,25], which is much better than the Aquarius mission goal; therefore, they can act as an in situ reference for the Aquarius mission.

The monthly 1° gridded Argo data used in this paper were generated from the float observations via the variational interpolation algorithm, which can be obtained from the Asia-Pacific Data-Research Center (APDRC) of the International Pacific Research Center (IPRC) at the University of Hawaii (http://apdrc.soest.hawaii.edu/projects/Argo/data/gridded/On_standard_levels/index-1.html). The averages of the two shallowest grid levels (0 and 5 m) from September 2011 to May 2015 were extracted in the present study for large-scale bias correction.

Individual Argo buoy observations from September 2011 to May 2015 were employed to validate the Aquarius gridded SSS fields. The individual Argo float measurements used in this paper were obtained from the Global Ocean Argo Scatter Dataset (V2.1) provided by the China Argo Real-time Data Center (CARDC) (ftp://ftp.argo.org.cn/pub/ARGO/global/). This dataset collects the observations of more than 12,900 autonomous profiling buoys placed in the global ocean by international Argo member states between January 1996 and May 2017. In addition, these observations have been strictly re-quality controlled by the CARDC. Only Argo profile measurements from each delay-mode Argo profile for depths shallower than 6 m and flagged as good were used in the verification process. The number of Argo float data per week is approximately 1000, and they present a quasirandom geographical distribution [13].

## 3. Method and Procedures

### 3.1. Satellite Bias Correction

Large-scale, long-term systematic biases are observed between the Aquarius SSS data and the gridded Argo SSS data. Their spatial distribution is characterized by an obvious zonality [26,27]. These biases differ among the three beams of the Aquarius radiometer. Differences between ascending and descending data are also evident, which are believed to be related to low-level radio frequency interference (RFI) from adjacent land areas [21]. Therefore, these large-scale biases must be corrected separately before the data are used. The large-scale SSS fields from Aquarius are constructed via the bin-weighted averaging of the raw Aquarius observations within 6° × 6° spatial bins; the weighting method is presented in Section 3.3. The large-scale biases can then be extracted by subtracting the gridded Argo SSS data from the large-scale Aquarius SSS fields. In this study, Aquarius observations from September 2011 to May 2015 were employed to construct the bias fields, which are shown in Figure 1. To remove the small-scale signals, which arise mainly from irregular sampling, the bias fields were smoothed with a two-dimensional running Hanning window with a half-width of 8° [14].

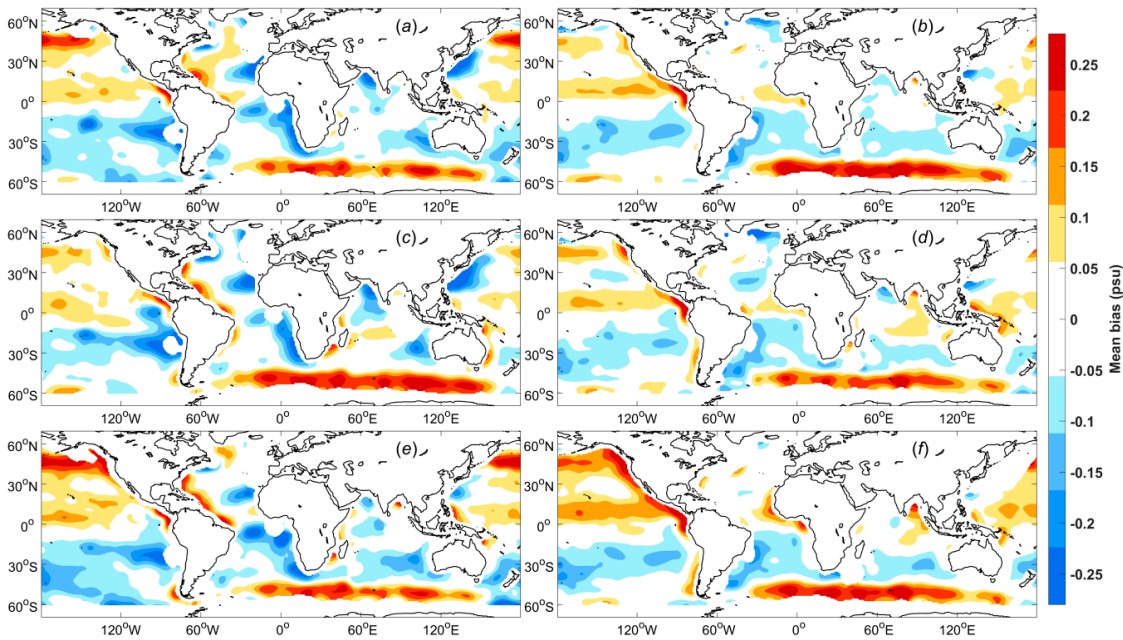

**Figure 1.** Mean spatial bias correction fields (psu) for Aquarius (left) ascending and (right) descending data: (top) beam 1, (middle) beam 2, and (bottom) beam 3.

Large positive biases are observed in high-latitude oceans, and large negative biases are observed in low-latitude oceans. In addition, the biases are different for ascending and descending satellite passes and for each of the three Aquarius beams (see Figure 1). By correcting these biases separately, the effects of RFI contamination and the residual interbeam biases can be effectively reduced [14].

The adjusted SSS ($S_{adj}$) can then be calculated as follows:

$$S_{adj} = S_{ret} - \Delta S \tag{1}$$

here, $S_{ret}$ is the retrieved value, and $\Delta S$ can be obtained from the bias fields, such as those shown in Figure 1.

### 3.2. Quality Control

The "radiometer_flags" dataset represents nonnominal data conditions detected for the radiometer measurements of each block and beam. At most, four numbers, which mostly represent different levels of data contamination, may be used to flag one SSS sample point. Each number is stored as an integer with 32 bits, where each bit represents a different data quality condition detected for that beam and block [5,15]. The values of any unused array elements are set to 0. Thus, a flag array **arr** with 4 rows and 32 columns can be built, as follows.

$$\mathbf{arr}_{4\times32} = \begin{pmatrix} 0 & 0 & 0 & 0 & 0 & 0 & 0 & 0 & 0 & 0 & 0 & 0 & 0 & 0 & 0 & 0 & 0 & 0 & 0 & 0 & 0 & 0 & 0 & 0 & 0 & \cdots & 0 \\ 0 & 0 & 1 & 1 & 0 & 0 & 1 & 0 & 0 & 0 & 0 & 1 & 0 & 0 & 1 & 0 & 0 & 0 & 1 & 0 & 0 & 0 & 0 & 0 & 0 & \cdots & 0 \\ 0 & 0 & 0 & 1 & 0 & 0 & 0 & 0 & 0 & 0 & 0 & 0 & 0 & 0 & 0 & 0 & 0 & 0 & 0 & 0 & 0 & 0 & 0 & 0 & 0 & \cdots & 0 \\ 0 & 0 & 0 & 0 & 0 & 0 & 1 & 0 & 0 & 0 & 0 & 1 & 0 & 0 & 0 & 0 & 0 & 0 & 0 & 0 & 0 & 0 & 0 & 0 & 0 & \cdots & 0 \end{pmatrix} \tag{2}$$

here, $arr(i, j)$ represents the $i$th member of flag bit $j$. An element value of 0 means that the data quality is good, while an element value of 1 means that the data quality is bad for the corresponding condition.

The L2 orbital SSS data should first be checked for quality to remove outliers during the process of producing the gridded products. The quality flags used for screening in this study, which are consistent with those used in the methods of Le Vine and Meissner [28] and Meissner [29], are shown

in Table 1. The corresponding Aquarius orbital SSS data were discarded whenever any of these flags was triggered.

**Table 1.** Quality flags used for screening.

| Condition Indicated | Last Flag Dimension | Element in arr |
|---|---|---|
| land contamination | severe, mask | $ar(1,3), arr(2,3)$ |
| sea ice contamination | severe, mask | $arr(1,4), arr(2,4)$ |
| wind/foam contamination | severe, mask, scat radio frequency interference (RFI) | $arr(1,5), arr(2,5), arr(3,5)$ |
| nonnominal navigation | roll, pitch, yaw, OOB | $arr(0,12), arr(1,12), arr(2,12), arr(3,12)$ |
| short accumulation overflow | overflow | $arr(0,13)$ |
| pointing anomaly | anomaly, asc_mode | $arr(0,16), arr(1,16)$ |
| rad Tb consistency | Tb_cons, emissivity | $arr(0,17), arr(1,17)$ |
| cold water | severe | $arr(1,18)$ |
| RFI level | severe | $arr(1,19)$ |
| nonnominal commanded state | mask | $arr(0,20)$ |
| reflected moon radiation | severe | $arr(1,21)$ |
| reflected galaxy radiation | severe | $arr(2,21), arr(3,21)$ |
| unacceptable Ascending/Dscending differences | mask | $arr(0,23)$ |

## 3.3. Quality and Distance Weighting

### 3.3.1. Weighting Method

The remaining data must be weighted before interpolation to distinguish measurements collected under different accuracy conditions. One important guideline for this weighting is to maximize the utilization of unflagged data and minimize unnecessary data loss.

In most previous studies, only the distance has been considered when determining the weight of each data point [30,31]. In fact, however, the data quality is also an important, nonnegligible factor. Therefore, in the method proposed here, both distance and data quality are considered in the weighting process, which can be expressed as follows:

$$w = w_{\text{qual}} \times w_{\text{dist}} \tag{3}$$

here, $w$ is the total weight, $w_{\text{qual}}$ is the quality weight and $w_{\text{dist}}$ is the distance weight.

There are several possible weighting functions, such as a parabolic function [32], a Gaussian weighting function and a tri-cubic kernel function [33]. However, the particular weighting functions does not appear to exert a particularly important effect on the solution performance [32]. Here, the Gaussian function is employed as the weighting function.

(1)  Quality weighting

The quality weight $w_{\text{qual}}$ can be calculated as follows:

$$w_{\text{qual}} = \exp\left(-k_1 \cdot x_{\text{qual}}^2\right) \tag{4}$$

here, $k_1$ is a constant, and $x_{\text{qual}}$ is a metric representing the quality of the data, which can be qualitatively determined as follows:

$$x_{qual} = \sum_{i=0}^{3} \sum_{j=0}^{31} arr(i,j) \tag{5}$$

In this qualitative weighting process, all elements in the array are assumed to have the same effect on the SSS; thus, all elements in **arr** have the same weight.

By means of an interpolation method (e.g., weighted average fitting), the SSS values can be acquired to generate the weekly gridded SSS fields. By comparing the gridded SSS data with the

concurrent individual Argo float SSS data, the RMSD, hereafter denoted by $R_0$, can be calculated to evaluate the influence of each condition on the accuracy of the gridded SSS fields.

In reality, however, different elements in the flag array do not have the same effect on the SSS. To account for these different effects, the individual elements also need to be weighted when calculating the data quality ($x_{\text{qual}}$). For any element $rr(m, n)$ ($0 \leq m \leq 3$, $0 \leq n \leq 31$), the corresponding quality can be written as the sum of all elements except $arr(m, n)$. The equation is as follows:

$$x_{qual,m,n} = \sum_{i=0}^{3} \sum_{j=0}^{31} arr(i, j) - arr(m, n) \tag{6}$$

Then, the weekly gridded SSS fields can be generated using the same interpolation method used for qualitative quality weighting, as described above. Again, by comparing the gridded SSS data with the concurrent individual Argo float SSS data, the corresponding RMSD, hereafter denoted by $R_{m,n}$ ($0 \leq m \leq 3$, $0 \leq n \leq 31$), can be calculated. The larger the difference between $R_{m,n}$ and $R_0$ is, the lower the reliability of $arr(m, n)$, which should be considered when calculating $w_{\text{qual}}$. Considering the Gaussian function shown in Equation (4), a less reliable $arr(m, n)$ should be given a greater weight when calculating $x_{\text{qual}}$. Simultaneously, the frequency of each condition should also be considered. The difference between $R_{m,n}$ and $R_0$ is the result of the combined action of all occurrences of the corresponding condition. Therefore, the weight $w_{m,n}$ of $arr(m, n)$ for a single occurrence can be calculated as follows:

$$w_{m,n} = (R_{m,n} - R_0)/f_{m,n} \tag{7}$$

Here, $f_{m,n}$ is the normalized frequency with which $arr(m, n)$ is flagged as 1.

After the weighting of all elements in **arr**, the calculation shown in Equation (5) can be further optimized to obtain a quantitatively weighted version of $x_{\text{qual}}$, as follows:

$$x_{\text{qual}} = k_2 \times \sum_{i=0}^{3} \sum_{j=0}^{31} \left[ arr(i, j) \times w_{i,j} \right] = k_2 \times \sum_{i=0}^{3} \sum_{j=0}^{31} \left[ arr(i, j) \times \left( R_{i,j} - R_0 \right)/f_{i,j} \right] \tag{8}$$

Here, $k_2$ is a constant.

(2) Distance weighting

The distance is also considered in the weighting process, in a manner similar to that used in the most recently proposed method [34]. The distance weight $w_{\text{dist}}$ can thus be defined as follows:

$$w_{\text{dist}} = \exp\left( -k_3 \times x_{\text{dist}}^2 \right) \tag{9}$$

Here, $k_3$ is a constant, and $x_{\text{dist}}$ is the Euclidean distance in kilometers.

### 3.3.2. Implementation

A statistical method was used here to determine the values of $k_1$, $k_2$ and $k_3$.

First, the constant $k_1$ was tuned. The value of $x_{\text{qual}}$ was calculated using only Equation (5), thus neglecting the influence of the distance weighting and the quantitative quality weighting. In other words, $k_2 = 0$ and $k_3 = 0$. In this way, the gridded SSS values were obtained using only qualitative quality weighting for different values of $k_1$. A series of RMSDs corresponding to the values of $k_1$ were calculated by comparing the gridded SSS data with the concurrent individual Argo float SSS data. The results reveal the dependence of the RMSD on the value of $k_1$, as shown in Figure 2a. The RMSD decreases rapidly with increasing $k_1$ in the range of 0 to 0.14. When $k_1$ is larger than 0.14, no evident differences in the RMSDs are observed. The minimum RMSD is 0.1965, which corresponds to $k_1 = 0.16$.

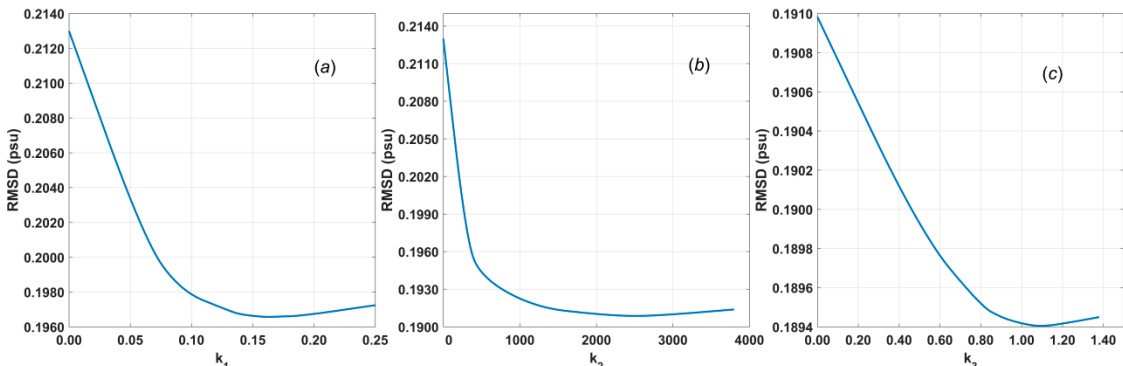

**Figure 2.** Relationships between the RMSD and (**a**) $k_1$, (**b**) $k_2$, and (**c**) $k_3$.

Then, the constant $k_2$ was tuned. The influence of the distance weighting on the SSS was again neglected, but the quantitative quality weighting was considered in this step. To this end, we let $k_1 = 0.16$ and $k_3 = 0$; then, the weight $w_{m,n}$ $(0 \leq m \leq 3, 0 \leq n \leq 31)$ of each element in **arr** was calculated using Equation (7). Only weights greater than $10^{-6}$ were retained. The conditions considered for quality weighting and the resulting weight values are listed in Table 2.

**Table 2.** Conditions considered for quality weighting and corresponding weight values.

| Condition indicated | Element | Weight ($10^{-4}$) | Frequency | $R_{m,n} - R_0$ ($10^{-5}$ psu) |
|---|---|---|---|---|
| (a) missing microwave radiometer (MWR) data | $w_{0,2}$ | 245.1 | 94181 | 301.0 |
| (b) moderate land contamination | $w_{0,3}$ | 65.0 | 108992 | 92.4 |
| (c) moderate sea ice contamination | $w_{0,4}$ | 7.0 | 48305 | 4.4 |
| (d) moderate wind/foam contamination | $w_{0,5}$ | 46.7 | 44970 | 27.4 |
| (e) V moderate unusual brightness temperature | $w_{0,6}$ | 149.2 | 86688 | 168.6 |
| (f) V severe unusual brightness temperature | $w_{1,6}$ | 14.9 | 18070 | 3.5 |
| (g) H moderate unusual brightness temperature | $w_{2,6}$ | 137.3 | 110336 | 197.6 |
| (h) H severe unusual brightness temperature | $w_{3,6}$ | 7.7 | 17992 | 1.8 |
| (i) V moderate sun glint | $w_{0,9}$ | 13.4 | 14308 | 2.5 |
| (j) V severe galactic contamination | $w_{1,11}$ | 171.8 | 50938 | 114.1 |
| (k) H severe galactic contamination | $w_{3,11}$ | 171.8 | 50937 | 114.1 |
| (l) roughness correction failure | $w_{1,14}$ | 1.1 | 14399 | 0.2 |
| (m) moderate cold water | $w_{0,18}$ | 2.8 | 52055 | 1.9 |
| (n) moderate RFI level | $w_{0,19}$ | 4.2 | 54726 | 3.0 |

Based on these results, Equation (8) can be rewritten as follows:

$$
\begin{aligned}
x_{\text{qual}} = \quad & k_2 \times arr(0,2) \times w_{0,2} + arr(0,3) \times w_{0,3} + arr(0,4) \times w_{0,4} + arr(0,5) \times w_{0,5} + arr(0,6) \times \\
& w_{0,6} + arr(1,6) \times w_{1,6} + arr(2,6) \times w_{2,6} + arr(3,6) \times w_{3,6} + arr(0,9) \times w_{0,9} + arr(1,11) \times \\
& w_{1,11} + arr(3,11) \times w_{3,11} + arr(1,14) \times w_{1,14} + arr(0,18) \times w_{0,18} + arr(0,19) \times w_{0,19}
\end{aligned}
\tag{10}
$$

The value of $x_{\text{qual}}$ was calculated using Equation (10). Then, the gridded SSS values with quantitative quality weighting were obtained for different values of $k_2$. A series of RMSDs corresponding to the values of $k_2$ were calculated by comparing the gridded SSS data with the concurrent individual Argo float SSS data. The results reveal the dependence of the RMSD on the value of $k_2$, as shown in Figure 2b. The RMSD decreases rapidly with increasing $k_2$ in the range of 0 to 1000. When $k_2$ is larger than 1000, no evident differences in the RMSDs are observed. The minimum RMSD is 0.1908, which corresponds to $k_2 = 2500$.

Finally, the constant $k_3$ was tuned. The influences of both the distance weighting and the quality weighting on the SSS values were considered in this step. To this end, we let $k_1 = 0.16$ and $k_2 = 2500$. The value of $x_{qual}$ was determined using Equation (10). Then, the gridded SSS values with distance weighting were obtained for different values of $k_3$. A series of RMSDs corresponding to the values of $k_3$ were calculated by comparing the gridded SSS data with the concurrent individual Argo float SSS data. The results reveal the dependence of the RMSD on the value of $k_3$, as shown in Figure 2c. The RMSD decreases rapidly with increasing $k_3$ in the range of 0 to 0.70. When $k_3$ is larger than 0.70, no evident differences in the RMSDs are observed. The minimum RMSD is 0.1894, which corresponds to $k_3 = 1.10$.

With $k_1 = 0.16$, $k_2 = 2500$ and $k_3 = 1.10$, the qualitative and quantitative quality weighting can reduce the RMSD by approximately 0.0165 and 0.0222, respectively, while the distance weighting can reduce the RMSD by approximately 0.0016, as indicated in Figure 2. The improvement in the RMSD achieved through quality weighting is approximately ten times that achieved through distance weighting. Thus, the accuracy of the traditional method of weighting the data according to distance can be significantly improved by including quality weighting.

*3.4. Interpolation Method*

Ideally, all available information should be extracted from the satellite data, ultimately yielding the optimal estimates of the evenly gridded SSS fields. Once the data have been weighted, three popular interpolation methods can be used to calculate the gridded SSS values: weighted average fitting, weighted unary linear fitting and weighted binary linear fitting.

(1)   Weighted average fitting (WAF)

Suppose that each $S_i$ $(1 \leq i \leq N)$ represents an observed salinity value and that N is the total number of observations. Then, the estimated salinity $\hat{S}$ can be calculated as follows:

$$\hat{S} = \frac{\sum_{i=1}^{N} S_i \times w_i}{\sum_{i=1}^{N} w_i} \tag{11}$$

Here, $w_i$ $(1 \leq i \leq N)$ is the weight of the *i*th observation, which depends on the spatial location and data quality.

(2)   Weighted unary linear fitting (WULF)

Let $x_i$ $(1 \leq i \leq N)$ be the distance from the *i*th observation to the central grid point; then, the estimated salinity $\hat{S}(x)$ can be calculated as follows:

$$\hat{S}(x) = a_0 + a_1 \times x \tag{12}$$

Here, $a_0$ and $a_1$ are regression coefficients.

For observation value $S_i$ observed at location $x_i$ $(1 \leq i \leq N)$, the WULF result is obtained by minimizing the following:

$$\sum_{i=1}^{N} |S_i - a_0 - a_1 \times x_i|^2 \times w_i \tag{13}$$

Here, $w_i$ $(1 \leq i \leq N)$ is the weight of the *i*th point, which depends on the spatial location and data quality.

(3)   Weighted binary linear fitting (WBLF)

Let $(x_i, y_i)$ $(1 \leq i \leq N)$ be the horizontal and vertical coordinates, with the center of the grid as the center of the coordinate system. The estimated salinity $\hat{S}(x, y)$ can be calculated via the following formula:

$$\hat{S}(x, y) = b_0 + b_1 \times x + b_2 \times y \tag{14}$$

Here, $b_0$, $b_1$ and $b_2$ are regression coefficients.

For data value $S_i$ observed at location $(x_i, y_i)$ $(1 \leq i \leq N)$, the WBLF result is obtained by minimizing the following:

$$\sum_{i=1}^{N} \left| S_i - b_0 - b_1 \times x_i - b_2 \times y_i \right|^2 \times w_i \tag{15}$$

Here, $w_i$ $(1 \leq i \leq N)$ is the weight of the $i$th point, which depends on the spatial location as well as the data quality.

## 4. Results and Verification

The weekly averaged near-global gridded SSS fields were calculated on a $0.25° \times 0.25°$ grid using the three methods introduced above for the period of September 2011 to May 2015. The search radius used in this study was 150 km. Figure 3a–c show the differences in the weekly Aquarius SSS fields constructed using the three interpolation methods for 2 July to 8 July 2013. For all three interpolation methods, the data were weighted in terms of both distance and data quality. Both WULF and WBLF are polynomial fitting methods, although these methods differ in the way they treat the independent variables. Figure 3d shows the Aquarius SSS L3 product data provided by ADPS with a $1° \times 1°$ spatial resolution.

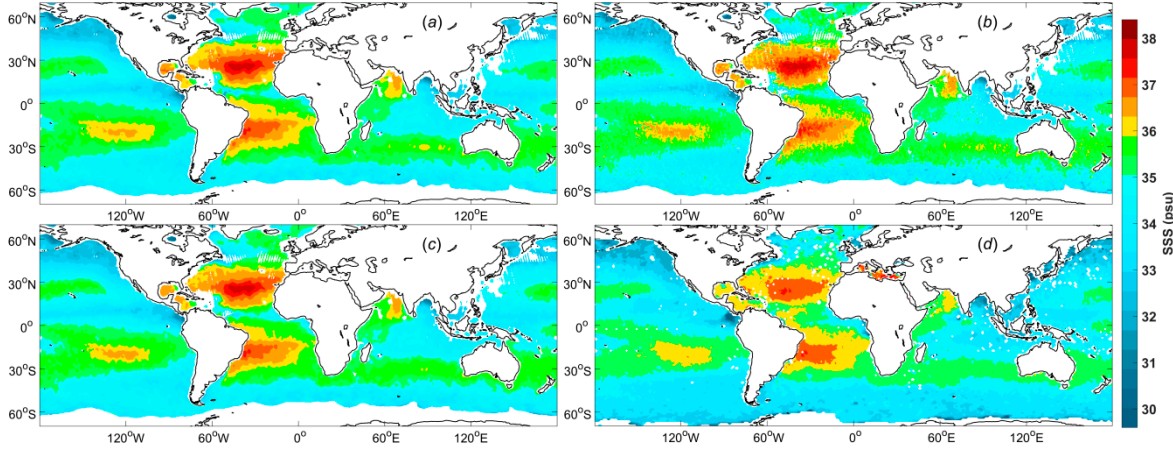

**Figure 3.** Weekly sea surface salinity (SSS) fields from Aquarius for 2 July–8 July 2013 constructed using different algorithms: (**a**) weighted average fitting (WAF), (**b**) weighted unary linear fitting (WULF), (**c**) weighted binary linear fitting (WBLF), and (**d**) the Aquarius SSS L3 product data provided by Aquarius Data Processing System (ADPS).

Intuitively, there are obvious differences in smoothness and in the SSS values. The values for WAF, WULF and WBLF are all higher than those of the Aquarius SSS L3 product data in most regions. Although both WBLF (Figure 3c) and WULF (Figure 3b) are polynomial fitting methods, there are notable differences between their SSS maps. The WULF SSS map exhibits a slight sawtooth phenomenon, while the WBLF SSS field is very smooth. This difference may arise because WBLF considers the spatial distribution of the data, while WULF does not. Moreover, the SSS field for WAF (Figure 3a) is slightly smoother than that for WBLF (Figure 3c).

Comparisons between the individual Argo measurements and the SSS values calculated with the three interpolation methods are shown in Figure 4. The magenta, green, blue and red curves are

time series of the weekly RMSDs (Figure 4a), biases (Figure 4b) and correlations (Figure 4c) of the results of WAF, WULF, and WBLF methods and the Aquarius L3 product data provided by ADPS. The analysis-to-buoy comparisons for the Aquarius L3 product data are presented here to answer the question of whether the new SSS fields are significantly more accurate than other Aquarius-derived SSS maps.

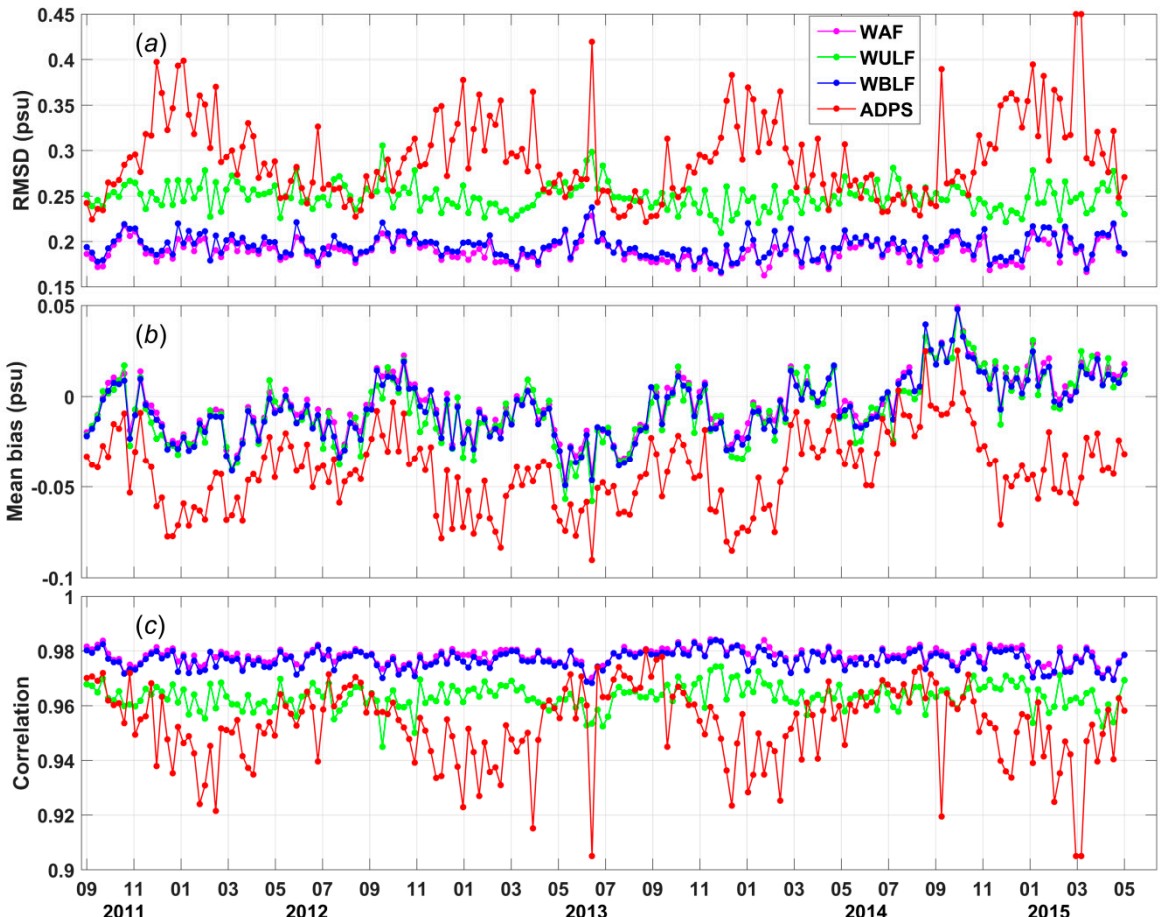

**Figure 4.** Time series of the weekly (**a**) root mean squared differences (RMSDs), (**b**) biases and (**c**) correlations between the Argo buoy data and the four Aquarius SSS analyses: WAF (magenta), weighted unary linear fitting (WULF) (green), WBLF (blue) and the official L3 SSS products provided by ADPS (red). The error statistics were calculated by comparing the Argo buoy measurements for a given week with the SSS values at the same locations obtained through interpolation of the corresponding SSS maps.

Significant differences are observed in the weekly RMSDs for the four analyses, as shown in Figure 4a. The weekly RMSDs for the WAF method vary from 0.1625 to 0.2283 psu. Similar to the WAF method, the weekly RMSDs for the WBLF method vary from 0.1665 to 0.2374 psu. The weekly RMSDs for both the WAF and WBLF methods are much better than those of the ADPS product data, which vary from 0.2212 to 0.4499 psu. The weekly RMSDs for the WAF and WBLF analyses are smaller than 0.2 psu for most weeks; specifically, this is true for 153 and 139 weeks, respectively, during the 194-week comparison period. The weekly RMSDs for the WULF analysis, which vary from 0.2093 to 0.3054 psu, are larger than the weekly RMSDs for the other two methods. However, the seasonal signal, which is obvious in the ADPS product data, is effectively reduced by all proposed methods.

As illustrated in Figure 4b, the WAF, WULF, and WBLF analyses can effectively reduce the biases but cannot completely eliminate them. These analyses may yield global bias time series that oscillate

around zero. By contrast, the global bias time series for the ADPS product data are almost entirely less than zero, implying that most of the ADPS SSS data are fresher than the Argo buoy data.

The weekly correlations of the WAF and WBLF results with the Argo measurements take values of approximately 0.98 for nearly all weeks (Figure 4c). The weekly correlation of the WULF results with the Argo measurements, which is approximately 0.96, is relatively small compared with those for the other two analyses. However, the weekly correlation of the ADPS product data with the Argo measurements is the worst, being as low as 0.9050 for some weeks.

Histograms of the differences between the Argo data and the four types of SSS data are shown in Figure 5. The percentages of the WAF SSS data (Figure 5a) and the WBLF SSS data (Figure 5c) that lie in the difference range from −0.1 psu to 0.1 psu are 48.97% and 48.32%, respectively; the corresponding percentage for the WULF SSS data is 39.91%, which is slightly smaller than that of 40.12% for the ADPS product data.

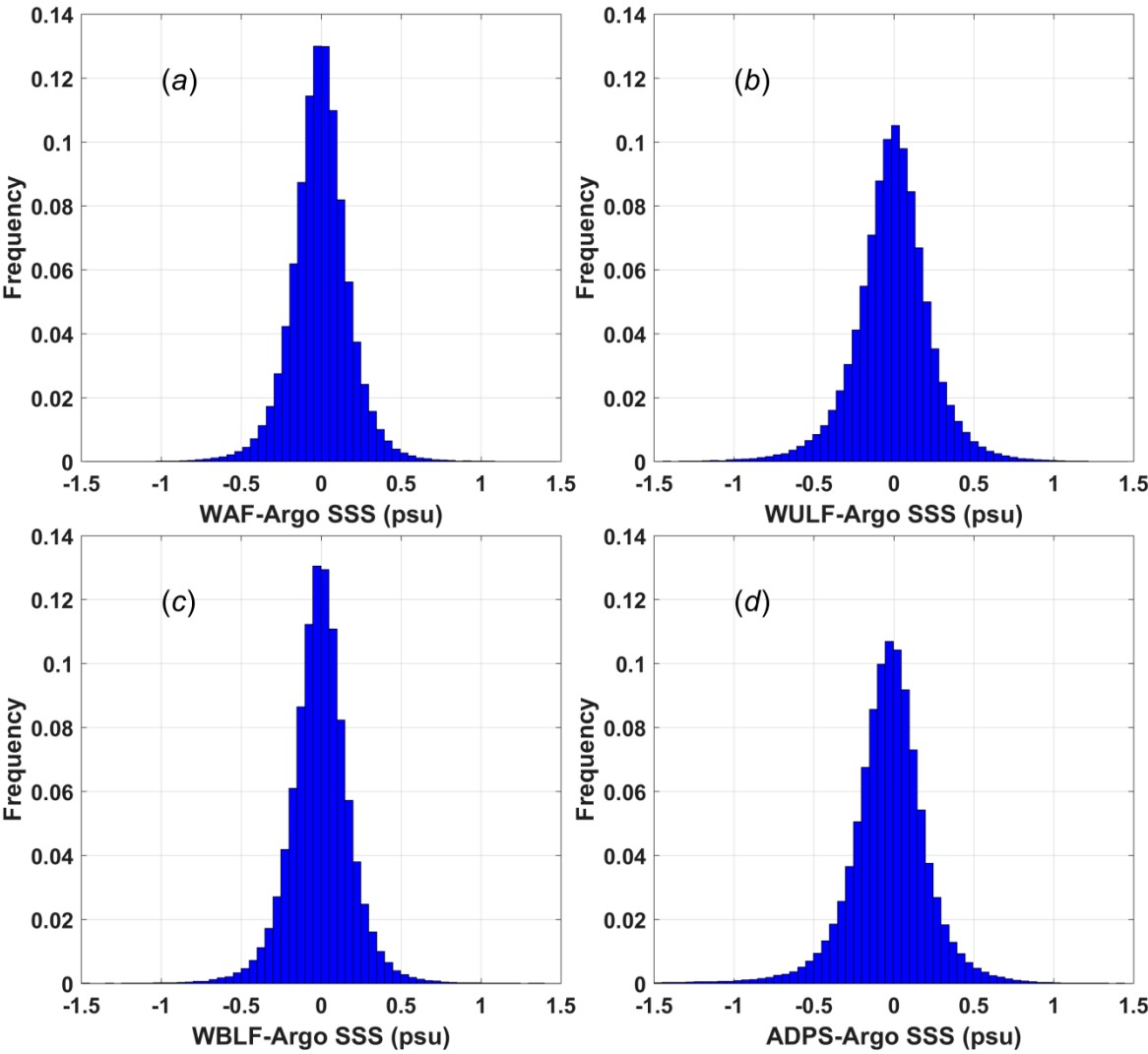

**Figure 5.** Statistics of the differences between the Argo buoy data and the results of the four Aquarius SSS analyses: (**a**) WAF, (**b**) WULF, (**c**) WBLF, and (**d**) the official standard L3 SSS products provided by ADPS. The error statistics were calculated by comparing the Argo buoy measurements for all weeks between September 2011 and May 2015 with the SSS values at the same locations obtained through interpolation of the corresponding Aquarius SSS maps.

The number of outliers (defined here as points with differences greater than 0.5 psu) is approximately 1.98% for the WAF SSS data, 2.15% for the WBLF SSS data, 4.08% for the WULF SSS data and 5.45% for the ADPS product data. Notably, the number of data points falling within ±0.1 psu for the ADPS product data is slightly larger than that for the WULF SSS data. However, the ADPS product data also feature a larger population of values with large differences (>0.5 psu), which can significantly reduce the accuracy of the gridded SSS fields. Therefore, the overall accuracy of the ADPS products is worse than those of the other three types of SSS data. Note that the mean and RMSD values are presented in Figure 6 below.

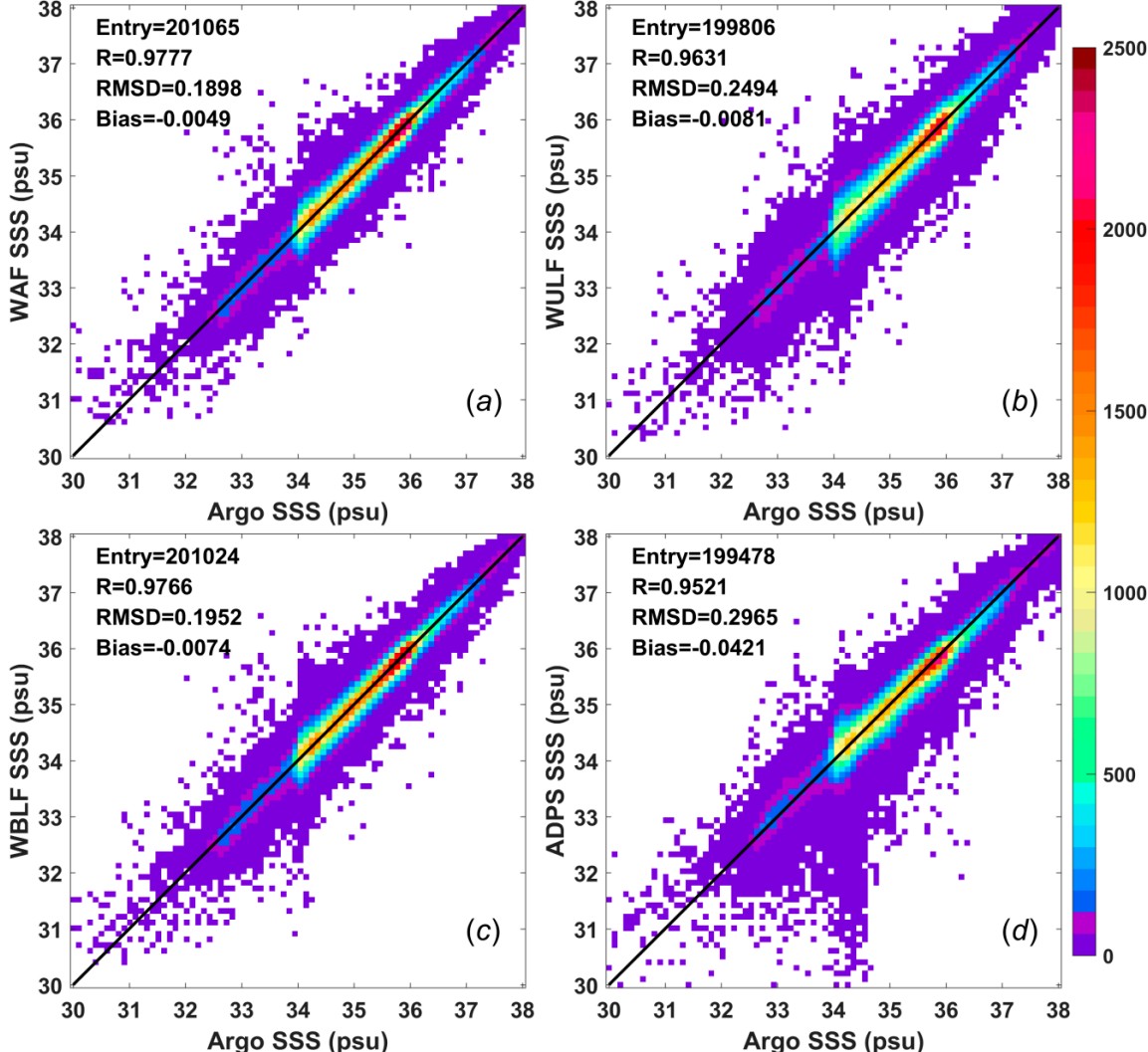

**Figure 6.** Scatter plots of the results of the Aquarius weekly SSS analyses and the collocated Argo buoy data. The Aquarius SSS analyses are (**a**) WAF, (**b**) WULF, (**c**) WBLF, and (**d**) the official standard L3 SSS product provided by ADPS. The colors represent the number of points in each 0.1 psu bin. The error statistics were calculated by comparing the Argo buoy measurements with the SSS values at the same locations obtained via interpolation of the corresponding Aquarius SSS maps for all weeks between September 2011 and May 2015.

Scatter plots of the results of the Aquarius weekly SSS analyses and the collocated Argo buoy data between September 2011 and May 2015 were generated to determine how well the actual data points matched the collocated Argo data (Figure 6). Theoretically, the numbers of collocated data, i.e., 200,065 for the WAF SSS analysis, 199,806 for the WULF SSS analysis, 200,024 for the WBLF SSS analysis and

199,478 for the ADPS product, should be the same. However, these values differ despite having the same duration because the numbers of missing values are different among the datasets.

The bias, which is defined as the Aquarius SSS value minus the Argo SSS value, is less than ±0.01 psu for all three analyses proposed in this paper. The mean bias for the WAF method, which is the smallest bias among all three proposed analyses, is −0.0049 psu, and this value is much smaller than the bias of the L3 product, namely, −0.0421 psu. These negative values imply that the SSS estimates from the Aquarius data are fresher than the Argo data, whereas a positive value would indicate the opposite. The region in Figure 6 with the highest density of points (red) for each of the WAF, WULF, and WBLF SSS analyses lies along the diagonal line. By contrast, the region with the highest density of points for the ADPS product is offset from the diagonal line, consistent with the relatively large negative bias; this pattern is also observed in Figure 4b.

The correlation coefficients among the datasets were also calculated (Figure 6). The plot of the WAF SSS data against the Argo SSS data shows that these two datasets have a very high correlation coefficient of 0.9777. The plot of the WBLF SSS data against the Argo SSS data exhibits a similar correlation coefficient of 0.9765. The plot of the WULF SSS data against the Argo SSS data yields a correlation coefficient of 0.9630, which is worse than those of the WAF and WBLF analyses but better than the correlation coefficient of 0.9521 found for the ADPS product. These results conform to the histograms shown in Figure 5.

Figure 6 also shows that the L3 product has the largest mean RMSD of 0.2965 psu, which is consistent with the time series of weekly RMSDs shown in Figure 4a. The mean RMSDs for the WAF and WBLF analyses are 0.1899 psu and 0.1952 psu, respectively, which are 0.1066 psu and 0.1013 psu smaller, respectively, than that for the ADPS product. The mean RMSD for the WULF analysis is 0.2495 psu, which is slightly smaller than that for the ADPS product.

## 5. Discussion

Among the interpolation methods, WAF is linear and the simplest method, whereas both WULF and WBLF are polynomial fitting methods. They all are widely used. The key differences between our methods and traditional methods lie in the weighting of the data. Although there are various weighting functions available, traditional methods treat only the distance as an independent variable, whereas we also consider the data quality. The improvement in the RMSD that is achieved through quality weighting is approximately ten times that achieved through distance weighting. Even when measurements with severe contamination have been discarded, moderate contamination may still exist. Therefore, results of higher accuracy can be obtained by assigning lower weights to data of lower quality. The results show that the accuracy of the output of our WAF method is improved by approximately 36% compared to the officially released weekly L3 products.

As shown in Table 2, 14 conditions were used for dual quality–distance weighting in this study, which are not considered in traditional methods. The weight of each element in **arr** (Table 2) is related to its total influence on the RMSD as well as the frequency and distribution of occurrence of the corresponding condition. Considering the Gaussian function shown in Equation (4), the larger the weight of an element is, the smaller its contribution to the final result, and the more effective the elimination of its effect. Note that the total influence on the RMSD, i.e., the difference between $R_{m,n}$ and $R_0$, may be very small because all conditions except the considered condition are still qualitatively weighted and thus have a dominant influence on the RMSD. The sum of the total influence on the RMSD exerted by all conditions should be equal to the difference between the RMSDs found with qualitative and quantitative quality weighting. The frequency of condition (g) (i.e., H moderate unusual brightness temperature) is the highest, at approximately 110336, whereas condition (l) (i.e., roughness correction failure) has the lowest frequency of 14399. Here, the frequency is calculated as the mean number of times that the corresponding element in **arr** is flagged as 1 within one week. The distributions for each condition are displayed in Figure 7, where conditions (a) to (n) correspond to those listed in Table 2.

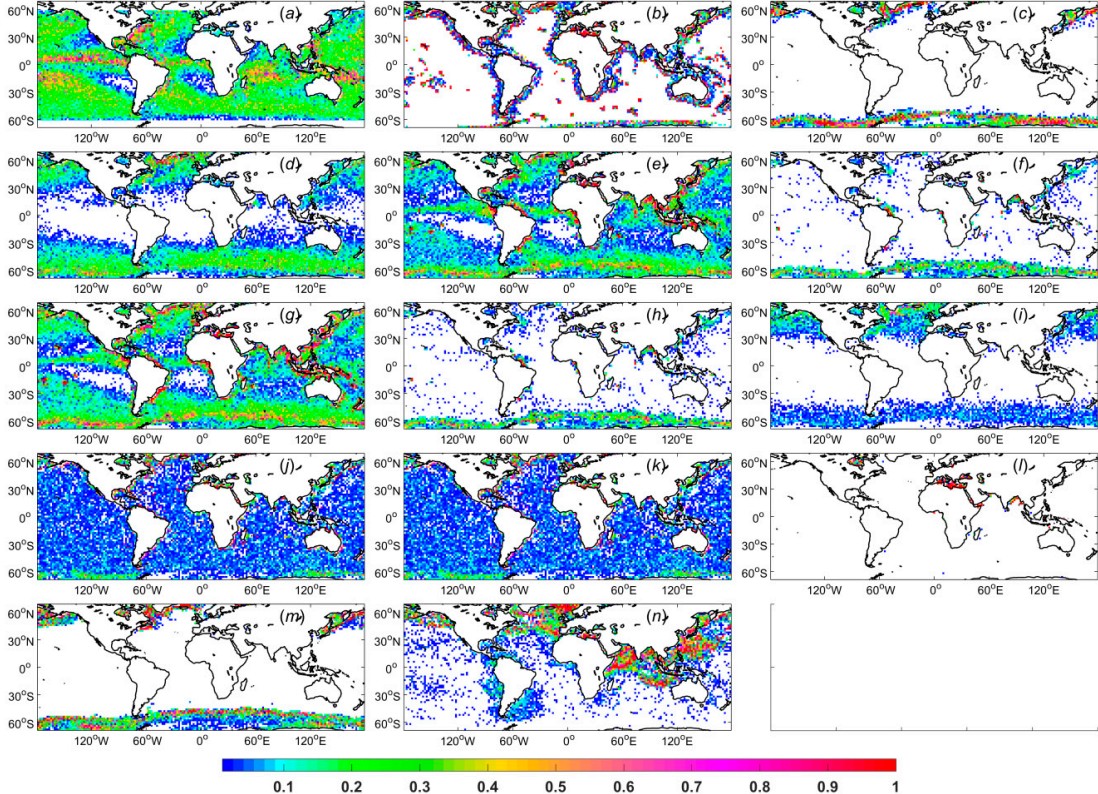

**Figure 7.** Distribution of each condition. The colors indicate the normalized frequency of each condition in each $2° \times 2°$ bin. The conditions represented in panels (**a**) to (**n**) directly correspond to those listed in Table 2.

It can be seen from Figure 7 and Table 2 that although the difference between $R_{m,n}$ and $R_0$ for (g) is slightly larger than that for (j), the weight of (g) is smaller than that of (j) because the frequency of (g) is much higher than that of (j). Meanwhile, although the frequencies of (j) and (m) are almost the same, the weight of (j) is far larger than that of (m). By considering their distributions, it can be found that (m) features an obvious regional distribution, whereas (j) does not. Therefore, we can conclude that in this case, the distribution dominates the weighting process. The effects of the conditions without obvious regional distributions, i.e., (a), (e), (g), (j), and (k), can be significantly reduced, whereas the conditions with obvious regional distributions, i.e., (b), (c), (l), and (m), cannot be weighted effectively. This finding implies that as long as unflagged data exist in most areas where flagged data appear, the impact of those flagged data can be effectively eliminated, and vice versa.

Many conditions, such as wind contamination, sea ice contamination and cold water, have been considered for screening in the process of validation and transformation from L2 to L3 [28,29]. However, there are some conditions, such as unusual brightness temperatures and galactic contamination, which have not been used in any previous processing methods but also have great impact on the results. Hence, the results of this study should serve to remind readers of the importance of considering these conditions.

A map of the mean SSS differences between the WAF and ADPS products (WAF–ADPS) for the period of September 2011 to May 2015 is shown in Figure 8. Obvious systematic biases are observed in the difference map, which are the result of the large-scale bias adjustment (Figure 8b) as well as the weighting process (Figure 8c). In particular, systematic differences are observed in many regions where salinity plays a critically important role in ocean dynamics. There are positive systematic biases in the South Temperate Zone, near near 30°S in the Pacific, Atlantic and Indian Oceans, which are mainly due to the large-scale bias adjustment. Negative systematic biases are observed in the North Pacific as

well as the regions around 50°S in the Atlantic and Indian Oceans, which are also mainly caused by the large-scale bias adjustment. Most regions near a coast are characterized by positive systematic biases. By considering the distributions for each condition (Figure 7), it can be found that for the five strongest influencing conditions, namely, (a), (j), (k), (e), and (g), high frequencies are also found in these near-coast regions. Because these regions show higher frequencies of these conditions, they should contribute less to the results than regions with lower frequencies do when weighted reasonably. Then, the results will be closer to the real values and show greater systematic biases with respect to the ADPS data, which do not include quality weighting. Positive systematic biases of approximately 0.2 psu are found in the Bay of Bengal, which are mainly caused by conditions (e), (g) and (n). There are also positive systematic biases of approximately 0.2 psu in the region near the Amazon River plume, which are mainly caused by conditions (a), (e), and (g). In the Labrador Sea, we can again see positive systematic biases of approximately 0.2 psu, which are mainly caused by conditions (b), (c), (g), (m), and (n). Negative systematic biases of approximately -0.2 psu are observed near 8°N in the Pacific Ocean, which are mainly caused by condition (a) in addition to the large-scale bias adjustment.

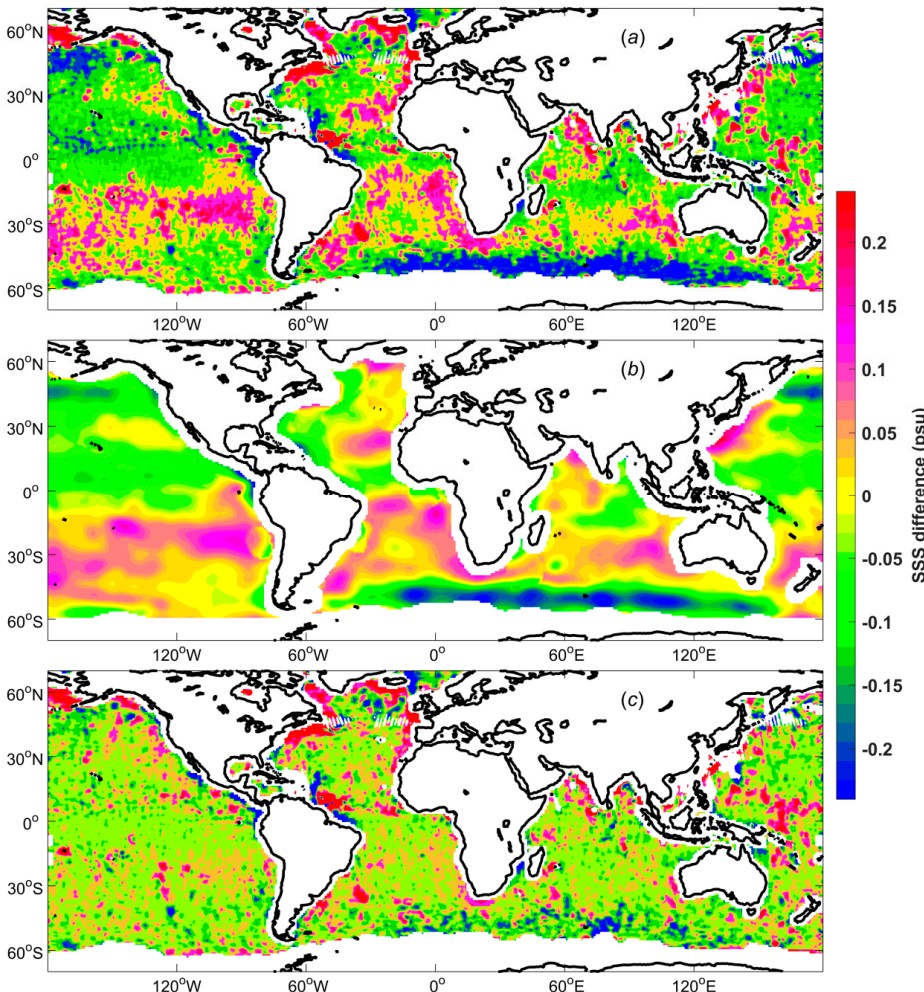

**Figure 8.** SSS difference map between the WAF and ADPS products (WAF - ADPS): (**a**) is the total difference, (**b**) is the part of the difference due to the large-scale bias adjustment, and (**c**) is the part of the difference due to the weighting process.

The Argo measurements should be treated as the ground truth. A previous study indicated that the salinity variation from the surface to a depth of 10 m is normally less than 0.1 psu, but when there is rain, the difference can be larger than 1 psu [35]. In this study, most data with large differences

were either screened out or reasonably weighted according to quality flags before being subjected to interpolation. When the Argo measurements are used to calculate the gridded fields, greater weights will be assigned to Aquarius SSS data whose values are closer to the Argo SSS values. Consequently, if the Argo SSS data are also used for validation, strong correlations may be found between the gridded SSS data and the Argo SSS data. Unfortunately, however, no adequate in situ data other than the Argo data are available. If possible, CTD or other in situ salinities should be used to calculate the gridded fields in future experiments.

The proposed methods have certain limitations. The weighting method proposed in this paper can be used only to eliminate the influence of conditions without obvious regional distributions. For a condition with an obvious regional distribution (such as cold water), the data cannot be effectively weighted. Moreover, a lack of coverage in the gridded SSS maps is observed in coastal regions and semi-enclosed seas, which mostly occurs because data with a large fraction of land contamination are discarded due to their low quality. In addition, a solid understanding of the physical basis of the proposed method is lacking because we have not found a unified standard for quantitatively evaluating the influence of different conditions. This shortcoming should be improved upon in future research.

Despite the aforementioned limitations, the results suggest that the proposed method shows good prospects in certain respects. First, the method can yield more accurate products at smaller spatial and temporal scales without small-scale noise. Second, the method can meet the demand for SSS analyses that are quantitatively consistent with existing in situ observations, such as those from Argo profile data. Additionally, the influences of different factors on salinity can be visualized and evaluated, thus serving as a reference for sensor design and reliability assessments of measurement results in the future.

## 6. Conclusions

A dual quality–distance weighting method is proposed in this paper. All types of nonnominal data conditions detected for the radiometer measurements of each block and beam are fully considered in our method. The key aspect of the method is the dual weighting of the data according to quality flags. By dual weighting the data in this way, the method can maximize the utilization of unflagged data while simultaneously minimizing unnecessary data loss.

The SSS satellite observations used in the study are officially released Aquarius L2 version 5.0 data. These data were first adjusted to eliminate large-scale systematic biases. In the weighting process, 14 data conditions were considered. Their geospatial distributions and influences on the SSS were visualized and evaluated. After the data were weighted, three popular interpolation methods, namely, WAF, WULF, and WBLF, were employed by setting a reasonable search radius. Weekly near-global gridded SSS fields were calculated on a $0.25° \times 0.25°$ grid using all three methods for the period of September 2011 to May 2015. In addition, the officially released weekly Aquarius L3 global mapped version 5.0 SSS products provided by ADPS were considered for comparison.

To verify the accuracy of the results, error statistics were calculated by comparing individual Argo measurements with the SSS values obtained for a given week at the same locations by interpolating the corresponding Aquarius SSS maps. The accuracy of the WAF SSS data was found to be very similar to that of the WBLF SSS data. The accuracy of the results of these two methods is improved by approximately 36% and 34%, respectively, compared to the officially released weekly L3 products. The accuracy of the WULF SSS results is relatively poor but is still improved by approximately 16%.

**Author Contributions:** Conceptualization, Y.L.; Methodology, Y.R.; Validation, Y.L.; Formal Analysis, Q.D.; Investigation, Y.R.; Resources, Y.L.; Writing—Original Draft Preparation, Y.L.; Writing—Review and Editing, Q.D. and Y.R.; Funding Acquisition, Q.D. and Y.R.

**Funding:** This work was supported in part by the National Key Research and Development Program of China (2017YFA0603003 and 2016YFC1401006), in part by a strategic pilot technology project (class A) of the Chinese Academy of Sciences (XDA19060504), in part by the National Natural Science Foundation of China (No. 41876210), and in part by a Key Science and Technology Planning Project of Hainan Province (ZDKJ2017006).

**Acknowledgments:** The authors thank the editors and anonymous reviewers for their comments, which have helped to improve the paper. Aquarius SSS data are available from https://oceandata.sci.gsfc.nasa.gov/Aquarius/, Argo monthly gridded data are available from http://apdrc.soest.hawaii.edu/projects/Argo/data/gridded/On_ standard_levels/index-1.html, and individual Argo float data collocated with Aquarius data are available from ftp://ftp.argo.org.cn/pub/ARGO/global/.

**Conflicts of Interest:** The authors declare no conflict of interest.

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
