# Peer review of "Aquarius Sea Surface Salinity Gridding Method Based on Dual Quality–Distance Weighting"

_remotesensing, doi:10.3390/rs11091131_

Round 1

Reviewer 1 Report

General structure, methodology and result analysis of this paper are clear. But the writing needs considerable improvements. There are many conflict meaning sentences in the context. Also, please examine all equations and parameter definitions. Detailed items from the review are listed below,

Using radiometer measurement quality flags as quantitative weighting is still a coarse approach. There are parameters in the Aquarius L2 data that present SSS error sources and uncertainty sensitivities. Have you considered comparing them with your weightings from equation (6). e.g. does weight value calculated for factor (b) in table 2, i.e. w(0,3) agree with the significance of land contamination in whole SSS uncertainty? or. compare w(0,3)/sum(w) with SSS_unc_TbV_land_contam/SSS_unc. This is just a rough idea. The uncertainty factors in L2 are not exactly the same as rad flags. line 427 statement is not quite true.

line 17: 36% compared to what?

line 32: "The official released..." should be "The officially released" Or "The official Aquarius gridded..." There are quite some grammar to be corrected in this paper. 

line 35-37: remove conclusions from old data, (V3.0); V5.0 was released Dec. 7, 2017

lin 54: "As a key dataset, quality flags known as “radiometer_flags” indicate that at least sixty nonnominal data conditions were detected for radiometer measurements" this sentence is confusing to readers without background of Aquarius radiometer calibration.  Try elaborating it or remove it.

line 76: "At most, four numbers are used to flag one SSS sample point. Each number is stored as an integer with 32 bits..." the way the rad flag data is organized in L2 is to match the format of vertical, plus 45, minus 45 and horizontal polarizations of each of the 3 radiometers. You can say the four numbers mostly represent different levels of data contamination. 

line 81: "An element value of 0 means that the data are unused or ..." confusing, or even conflict meanning

consider moving section 2.4 before 2.2 to give readers a clear idea of how you are grinding the data. or at least a brief description of what you are doing with the weight factors

line 99: equation (2), W-qual = w1 * w2 from "Quality qualitative weighting" and "Quality quantitative weighting" respectively? or not?

line 105: The whole part of "Quality qualitative weighting" in section 2.2.1 needs careful re-writing. equation (3) doesn't match equation (4), and which is not conceptually consistent with equation (7).

line 134: what is the unit used for euclidean distance, more details

section 2.2.2, have you considered computing k3 first by throwing away all flagged data? then compute k1 and k2?

line 151: table 2, (a) missing MWR, RIM has been used to compute rain effects since V4.0; also, it is better to exclude measurements with rain from the Argo match-up due to the stratification

one major concern of this method, some factors in table 2 are correlated, counting all the weighting together in equation (9) might over emphasize the correlating factors.

Fig. 5, what are the mean, and std values of each panel?

line 344: figure 4b?

line 385, (j) is not random, the galactic effect is a predictable pattern; (n) is rather more random

last sentence in line 410 seems to have conflict meaning

figure 7, it might be more meaningful to normalize the "frequency"

fig 8, how is the diff computed? ADPS - WAF? what is the time period of data set used?

Author Response

Response to Reviewer 1 Comments

We would like to thank the reviewers for their detailed and helpful suggestions, comments and careful checking. Below, we respond on a point-by-point basis. Please see our revised manuscript for details.

Comment: General structure, methodology and result analysis of this paper are clear. But the writing needs considerable improvements. There are many conflict meaning sentences in the context. Also, please examine all equations and parameter definitions. Detailed items from the review are listed below,

R: We thank the reviewer for the positive comments. We have tried to improve the manuscript to make it read better and clarify several points.

Q1: Using radiometer measurement quality flags as quantitative weighting is still a coarse approach. There are parameters in the Aquarius L2 data that present SSS error sources and uncertainty sensitivities. Have you considered comparing them with your weightings from equation (6). e.g. does weight value calculated for factor (b) in table 2, i.e. w(0,3) agree with the significance of land contamination in whole SSS uncertainty? or. compare w(0,3)/sum(w) with SSS_unc_TbV_land_contam/SSS_unc. This is just a rough idea. The uncertainty factors in L2 are not exactly the same as rad flags. line 427 statement is not quite true.

R1: Thank you for your advice. This is a good idea that we had not previously considered. We will address it in future research.

Q2: line 17: 36% compared to what?

R: This improvement is calculated in comparison to the officially released weekly level 3 products. We have specified this in the manuscript.

Q3: line 32: "The official released..." should be "The officially released" Or "The official Aquarius gridded..." There are quite some grammar to be corrected in this paper. 

R3: We apologize for this mistake. We have corrected it.

Q4: line 35-37: remove conclusions from old data, (V3.0); V5.0 was released Dec. 7, 2017

R4: We have removed these conclusions.

Q5: line 54: "As a key dataset, quality flags known as “radiometer_flags” indicate that at least sixty nonnominal data conditions were detected for radiometer measurements" this sentence is confusing to readers without background of Aquarius radiometer calibration.  Try elaborating it or remove it.

R5: We have modified this sentence.

Q6: line 76: "At most, four numbers are used to flag one SSS sample point. Each number is stored as an integer with 32 bits..." the way the rad flag data is organized in L2 is to match the format of vertical, plus 45, minus 45 and horizontal polarizations of each of the 3 radiometers. You can say the four numbers mostly represent different levels of data contamination. 

R6: We have made this modification. Thank you for your advice.

Q7: line 81: "An element value of 0 means that the data are unused or ..." confusing, or even conflict meaning

R7: We have revised this sentence.

Q8: consider moving section 2.4 before 2.2 to give readers a clear idea of how you are grinding the data. or at least a brief description of what you are doing with the weight factors

R8: We have made adjustments to the structure of section 2.

Q9: line 99: equation (2), W-qual = w1 * w2 from "Quality qualitative weighting" and "Quality quantitative weighting" respectively? or not?

R9: No, it is not. We did not make it clear. W-qual can be calculated either with "qualitative quality weighting" or with "quantitative quality weighting". "Quantitative quality weighting" is a further optimization of "qualitative quality weighting". We have revised the manuscript to clarify this.

Q10: line 105: The whole part of "Quality qualitative weighting" in section 2.2.1 needs careful re-writing. equation (3) doesn't match equation (4), and which is not conceptually consistent with equation (7).

R10: We apologize for our carelessness. We have made the necessary corrections.

Q11: line 134: what is the unit used for euclidean distance, more details

R11: The unit is kilometers. We have added it.

Q12:section 2.2.2, have you considered computing k3 first by throwing away all flagged data? then compute k1 and k2?

R12: Yes, we have considered attempting to compute k3 first, but the result was not very good, mainly with respect to the effectiveness of the distance weighting. As mentioned in the manuscript, the reduction in the RMSD achieved through distance weighting was only approximately one-tenth of the reduction achieved through quality weighting. When k3 was computed first, the reduction in the RMSD achieved through distance weighting was even smaller.

Q13:line 151: table 2, (a) missing MWR, RIM has been used to compute rain effects since V4.0; also, it is better to exclude measurements with rain from the Argo match-up due to the stratification

R13: We have attempted to exclude rain rates higher than 0.25 mm/hr (arr(1,2)). However, no data were available for regions above 60 degrees in latitude when we excluded these rainy data. Therefore, to include more salinity information at high latitudes, these data were retained. However, because there were few Argo measurements available at high latitudes, the Aquarius data affected by rain could not be effectively weighted.

Q14: one major concern of this method, some factors in table 2 are correlated, counting all the weighting together in equation (9) might over emphasize the correlating factors.

R14: That indeed a problem. However, the most significant effect on W_qual is the exponential function. As the value of x_qual increases, the sensitivity of W_qual to x_qual gradually decreases and eventually becomes saturated. Therefore, this overemphasis in x_qual may ultimately result in little change in W_qual. Of course, this is only a preliminary guess. In our future research, we will try to find more reasonable weighting methods if necessary.

Q15: Fig. 5, what are the mean, and std values of each panel?

R15: The mean and RMSD values in Fig. 5 are shown in Fig. 6.

Q16: line 344: figure 4b?

R16: No, this sentence refers to the statistics in Fig. 6.

Q17: line 385, (j) is not random, the galactic effect is a predictable pattern; (n) is rather more random

R17: We have made appropriate modifications. Thank you for pointing this out.

Q18: last sentence in line 410 seems to have conflict meaning

R18: We have corrected this sentence.

Q19: figure 7, it might be more meaningful to normalize the "frequency"

R19: We have made the appropriate modification.

Q20: fig 8, how is the diff computed? ADPS - WAF? what is the time period of data set used?

R20: The difference is computed as WAF - ADPS, and the time period of the dataset used is from September 2011 to May 2015. The information has been added to the manuscript.

Reviewer 2 Report

The manuscript “Aquarius Sea Surface Salinity Gridding Method Based on Quality Dual Weighting” is within the scope of the journal and addresses a topic of interest to a large audience.

The abstract should be more informative. Please include more quantitative data and a better description of the work.

The introduction is somewhat unbalanced. In some parts of it offer a detailed and mature overview. In others, especially from line 54 onwards its quality decays and should therefore be revised.

More information on how the values were obtained for Figure 2 is needed.

Overall the manuscript is well written and the analysis sound and reproducible. Section 2 of the manuscript could be revised. The sub-divisions are not very coherent and break the readability of the text.

It is also recommended that the authors focus somewhat more in the advantages of the method when compared with other attempts of doing the same, which is not clear. The discussion of the drivers of the improvement of the new outputs (when compared to the reference set) is also lacking some depth. This analysis is too descriptive and lacks a careful consideration and discussion of the potential sources of error (including regional analyses, as there is a clear regionalization of the error).

Nonetheless, the manuscript warrants publication upon the minor changes recommended.

Author Response

Response to Reviewer 2 Comments

We would like to thank the reviewers for their detailed and helpful suggestions, comments and careful checking. Below, we respond on a point-by-point basis. Please see our revised manuscript for details.

Q1: The abstract should be more informative. Please include more quantitative data and a better description of the work.

R1: Thank you for pointing out this shortcoming. We have made appropriate modifications.

Q2: The introduction is somewhat unbalanced. In some parts of it offer a detailed and mature overview. In others, especially from line 54 onwards its quality decays and should therefore be revised.

R2: The introduction has been improved.

Q3: More information on how the values were obtained for Figure 2 is needed.

R3: We have added this information in section 3.1.

Q4: Overall the manuscript is well written and the analysis sound and reproducible. Section 2 of the manuscript could be revised. The sub-divisions are not very coherent and break the readability of the text.

R4: We have made adjustments to the structure of section 2.

Q5:It is also recommended that the authors focus somewhat more in the advantages of the method when compared with other attempts of doing the same, which is not clear.The discussion of the drivers of the improvement of the new outputs (when compared to the reference set) is also lacking some depth.

R5: Thank you for pointing this out. We have added to the discussion in section 5. Please refer to the revised manuscript.

Q6:This analysis is too descriptive and lacks a careful consideration and discussion of the potential sources of error (including regional analyses, as there is a clear regionalization of the error).

R6: We have added a further discussion of the potential sources of error in section 5. More detailed analyses will be done in the future.

Comment: Nonetheless, the manuscript warrants publication upon the minor changes recommended.

R: We thank the reviewer for these positive comments. We have tried to improve the manuscript to make it read better and clarify several points.
